# Chronic Intermittent Hypoxia-Induced Dysmetabolism Is Associated with Hepatic Oxidative Stress, Mitochondrial Dysfunction and Inflammation

**DOI:** 10.3390/antiox12111910

**Published:** 2023-10-25

**Authors:** Joana L. Fernandes, Fátima O. Martins, Elena Olea, Jesus Prieto-Lloret, Patrícia C. Braga, Joana F. Sacramento, Catarina O. Sequeira, Ana P. Negrinho, Sofia A. Pereira, Marco G. Alves, Asunción Rocher, Silvia V. Conde

**Affiliations:** 1iNOVA4Health, NOVA Medical School, Faculdade de Ciências Médicas, Universidade NOVA de Lisboa, 1150-069 Lisboa, Portugal; joana.fernandes@nms.unl.pt (J.L.F.); fatima.martins@nms.unl.pt (F.O.M.); joana.sacramento@nms.unl.pt (J.F.S.); catarina.sequeira@nms.unl.pt (C.O.S.); anapatricia.negrinho@nms.unl.pt (A.P.N.); sofia.pereira@nms.unl.pt (S.A.P.); 2Departamento de Enfermeria, Universidad de Valladolid, 47005 Valladolid, Spain; elena.olea@uva.es; 3Unidad de Excelencia Instituto de Biomedicina y Genética Molecular (IBGM), Consejo Superior de Investigaciones Científicas, Universidad de Valladolid, 47005 Valladolid, Spain; jesus.prieto@uva.es (J.P.-L.); rocher@ibgm.uva.es (A.R.); 4Departamento de Bioquímica, Biologia Molecular y Fisiologia, Universidad de Valladolid, 47005 Valladolid, Spain; 5Unit for Multidisciplinary Research in Biomedicine (UMIB), Institute of Biomedical Sciences Abel Salazar (ICBAS), University of Porto, 4050-313 Porto, Portugal; patriciacbraga.1096@gmail.com (P.C.B.); alvesmarc@gmail.com (M.G.A.); 6ITR-Laboratory for Integrative and Translational Research in Population Health, 4050-313 Porto, Portugal; 7Institute of Biomedicine—iBiMED and Department of Medical Sciences, University of Aveiro, 3810-193 Aveiro, Portugal

**Keywords:** obstructive sleep apnea, chronic intermittent hypoxia, insulin resistance, metabolic disorders, mitochondrial dysfunction, inflammation, oxidative stress

## Abstract

The association between obstructive sleep apnea (OSA) and metabolic disorders is well-established; however, the underlying mechanisms that elucidate this relationship remain incompletely understood. Since the liver is a major organ in the maintenance of metabolic homeostasis, we hypothesize that liver dysfunction plays a crucial role in the pathogenesis of metabolic dysfunction associated with obstructive sleep apnea (OSA). Herein, we explored the underlying mechanisms of this association within the liver. Experiments were performed in male Wistar rats fed with a control or high fat (HF) diet (60% lipid-rich) for 12 weeks. Half of the groups were exposed to chronic intermittent hypoxia (CIH) (30 hypoxic (5% O_2_) cycles, 8 h/day) that mimics OSA, in the last 15 days. Insulin sensitivity and glucose tolerance were assessed. Liver samples were collected for evaluation of lipid deposition, insulin signaling, glucose homeostasis, hypoxia, oxidative stress, antioxidant defenses, mitochondrial biogenesis and inflammation. Both the CIH and HF diet induced dysmetabolism, a state not aggravated in animals submitted to HF plus CIH. CIH aggravates hepatic lipid deposition in obese animals. Hypoxia-inducible factors levels were altered by these stimuli. CIH decreased the levels of oxidative phosphorylation complexes in both groups and the levels of SOD-1. The HF diet reduced mitochondrial density and hepatic antioxidant capacity. The CIH and HF diet produced alterations in cysteine-related thiols and pro-inflammatory markers. The results obtained suggest that hepatic mitochondrial dysfunction and oxidative stress, leading to inflammation, may be significant factors contributing to the development of dysmetabolism associated with OSA.

## 1. Introduction

According to the American Academy of Sleep Medicine, sleep-related breathing disorders are a group of conditions defined by irregularities in respiration during sleep [1]. These diseases are subdivided into four classes: obstructive sleep apnea syndromes, sleep-related hypoventilation disorders, central apnea syndromes, and sleep-related hypoxemia disorder [2]. Obstructive sleep apnea (OSA) is the most common sleep-related breathing disorder, affecting nearly 1 billion adults aged 30–69 that were mostly men.

OSA is characterized by repetitive episodes of reduction (hypoapnea) or complete (apnea) cessation of the airflow due to the collapse of the pharyngeal airway during sleep [1]. These events promote a decrease in oxygen saturation and hypercapnia, which stimulate chemoreceptors and activate the sympathetic nervous system (SNS) to restore oxygen levels [3,4]. Chronic intermittent hypoxia (CIH) is the result of these constant apnea-hypoapnea cycles, and the continued respiratory-related arousals result in sleep fragmentation (SF) [4].

Over the years, the link between OSA and its associated comorbidities—cardiovascular disease, metabolic disorders, neurological and pulmonary diseases—has been studied to understand the pathophysiological mechanisms behind these relations and to develop new forms of treatment. OSA, in particular CIH and SF, have been correlated with different metabolic disorders, including obesity, metabolic syndrome (MetS) and type 2 diabetes (T2D) [5,6]. The association between OSA and dysmetabolism have been demonstrated in different clinical and epidemiological studies. MetS is extremely prevalent in OSA patients, with a reported variable rate of up to 80% [7]. Recently, it was demonstrated that at least 50% of T2D patients also have OSA, this disorder also being extremely prevalent in obese patients [8]. Nevertheless, this association was also observed in non-obese patients, highlighting that this relationship is independent of obesity [8]. Moreover, the severity of OSA positively correlate with the degree of insulin resistance and glucose intolerance, two pathological conditions that increase the risk of developing T2D [6,8]; therefore, OSA is considered an independent risk factor for T2D [6,8]. Also, there is a bidirectional relationship between these two disorders, with T2D being a risk factor for developing OSA [5]. Taken together, there is a clear association between the CIH and SF present in OSA and MetS and its core components; however, the mechanistic alterations connecting these disorders remain to be proven [5,7,9]. Another MetS-related disorder is NAFLD, which is the most prevalent chronic liver disease affecting approximately 30% of the adult population, being more prevalent in men [10,11]. Other factors that influence the prevalence of this disorder include age and the presence of other pathologies such as OSA [12]. 

At a molecular level, different pathophysiological mechanisms were proposed to explain the link CIH–dysmetabolism [5,13,14,15], including increased sympathetic activation, dysregulation of the hypothalamus–pituitary axis, alterations in adipokine levels’ increase in mitochondrial dysfunction, oxidative stress and reactive oxygen species (ROS) production, inflammation, and expression of genes involved in lipogenesis, among others [6]. Recently, we investigated whether adipose tissue dysfunction could be the main trigger of metabolic dysfunction in CIH; however, we found that CIH induced early-stage metabolic dysfunction, characterized by hyperinsulinemia and whole-body insulin resistance and without alterations in weight gain, in adipocytes’ perimeter, and in adipose tissue hypoxia, angiogenesis, oxidative stress, and metabolism [16].

As the liver plays a significant role in maintaining metabolic homeostasis, herein we hypothesized that liver dysfunction is a crucial factor in the pathogenesis of metabolic dysfunction associated with OSA. We further explored the underlying mechanisms of this association within the liver. We observed, as previously described, that CIH promotes the development of insulin resistance and glucose intolerance in control and high-fat fed animals, strengthening the already established link between OSA and dysmetabolism. Additionally, we demonstrated that CIH impacts the liver by increasing lipid deposition and inflammation and by decreasing the levels of oxidate phosphorylation complexes (OXPHOS), without changing the high-fat diet-induced alterations on hepatic mitochondrial density and antioxidant status.

Altogether, these results confirm the link between OSA and dysmetabolism, suggesting that mitochondrial dysfunction and oxidative stress, along with the consequent onset of inflammation, might be key factors that contribute to the development of this CIH-induced dysmetabolic phenotype.

## 2. Materials and Methods

### 2.1. Animals and In Vivo Procedures

Experiments were performed in male Wistar rats, aged 12 weeks, and obtained from the vivarium of the Faculty of Medicine of the University of Valladolid, Spain. The animals were randomly divided into two groups: a control group of 16 animals (CTL), fed with a standard control diet—3.8 kcal/g with 10% kcal as fat (D12450B; Open Source Diets)—and an obese group of 14 animals (HF), fed with a high-fat diet (HF diet)—5.2 kcal/g with 60% kcal as fat (D12492; Open Source Diets)—for 12 weeks. During the experimental period, the animals had food and water ad libitum and were kept under temperature and humidity control, with a regular light (08:00–20:00 h) and dark (20:00–08:00 h) cycle. In the last 2 weeks of the diet, half of the animals from each group were submitted to a CIH protocol (Figure 1). For that, animals were housed hermetically shielding in transparent methacrylate chambers (16 L, a maximum of four rats per chamber), with ad libitum access to food and water, as previously described by Gonzalez-Martín et al. [17]. The CIH protocol consisted of 30 cycles/h each cycle with an exposure for 40 s to 5% O_2_, followed by an exposure for 80 s to air, for 8 h per day (apnoea-hypopnoea index of 30). During the CIH protocol, the remaining animals, either from the control or obese group, were kept in the same room but were only exposed to a normal air atmosphere. Note that the PaO_2_ in animals during each cycle of 5% O_2_ drops to approximately 20 mmHg without changes in PaCO_2_. Also, note that the drop in PaO_2_ or the PaCO_2_ does not change at the beginning of the CIH treatment and after 15 days of CIH. After the total 12 weeks of the experimental period, an intraperitoneal glucose tolerance test was performed as previously described [18], and blood was collected for assessment of insulin sensitivity through the homeostasis model assessment (HOMA-IR) index. Animals were sacrificed by an intracardiac overdose of sodium pentobarbital, and the liver was immediately collected and placed in liquid nitrogen or in a 4% paraformaldehyde (PFA) solution. Samples were stored at −80 °C or at 4 °C, respectively, for posterior use.

Laboratory care was carried out following the European Union Directive for Protection of Vertebrates Used for Experimental and Other Scientific Ends (2010/63/EU). Experimental protocols were approved by the NOVA Medical School, Faculdade de Ciências Médicas and Valladolid University Ethics Committee.

### 2.2. Assessment of the Levels of Key Proteins Involved in Insulin Signaling, Glucose Metabolism, Mitochondrial Biogenesis, Antioxidant System, and Inflammation Pathways

Liver tissue was homogenized in a lysis buffer containing a protease inhibitor cocktail, as previously described [19]. Samples containing 20 μg of total protein were denatured in a dry bath for 5 min at 99 °C for all proteins, except for oxidative phosphorylation complexes’ (OXPHOS) analysis, and were denatured for 10 min at 50 °C. Afterwards, samples were separated by SDS-PAGE (10%) and transferred to a nitrocellulose membrane, as previously described [18]. After blocking in 5% non-fat milk in Tris-buffered saline with a 0.1% Tween (TBS-T 0.1%) solution for 1.5 h at room temperature, the membranes were incubated overnight at 4 °C with the following primary antibodies: HIF-1α (1:200, Sicgen, Cantanhede, Portugal), HIF-2α (1:200 Abcam, Cambridge, UK), insulin receptor (IR; 1:100, Santa Cruz Biotechnology, Madrid, Spain), protein kinase B (Akt) (1:1000, Cell Signaling Technologies, Madrid, Spain), insulin degrading enzyme (IDE) (1:1000, Abcam, Cambridge, UK), glucose transporter type 2 (Glut2) (1:500, Santa Cruz Biotechnology, Madrid, Spain), OXPHOS (1:500, MitoSciences, Eugene, OR, USA), superoxide dismutase 1 (SOD-1) (1:500, Santa Cruz Biotechnology, Madrid, Spain), catalase (1:500, SicGen, Cantanhede, Portugal), arginase I (1:500, Santa Cruz Biotechnology, Madrid, Spain), interleukin 6 receptor (IL6R) (1:200, Santa Cruz Biotechnology, Madrid, Spain), interleukin 1 receptor (IL1R) (1:100, Santa Cruz Biotechnology, Madrid, Spain), tumor necrosis factor alfa receptor (TNF- αR) (1:200, Santa Cruz Biotechnology, Madrid, Spain), tumor necrosis factor alfa (TNF- α) (1:1000, SicGen, Cantanhede, Portugal), and nuclear factor kappa beta (NF-kB) (1:500, Cell Signaling Technologies, Madrid, Spain). After washing with TBS-T 0.1%, membranes were incubated with goat anti-mouse (1:1000, Bio-Rad, Madrid, Spain), goat anti-rabbit (1:1000, Bio-Rad, Madrid, Spain), or mouse anti-goat (1:2000, Santa Cruz Biotechnology, Dallas, TX, USA) for 1.5 h at room temperature.

Membranes were then developed with enhanced chemiluminescence reagents according to the manufacturer’s instructions (Clarity Western ECL Substrate, BioRad, Madrid, Spain). The intensity of the signals was detected in a Chemidoc Molecular Imager (Chemidoc, BioRad, Madrid, Spain) and quantified using the Image Lab software version 6.1 (Bio-Rad, Hercules, CA, USA). The membranes were re-probed and tested to calnexin (1:1000, SicGen, Cantanhede, Portugal) or β-actin (1:1000, SicGen, Cantanhede, Portugal) to compare and normalize the protein expression with the amount of protein loaded. The mean intensity of control samples in each membrane was used as reference for controlling gel-to-gel variation.

### 2.3. Histological Analysis of Lipid Deposition and Immunohistochemistry Analysis of Hypoxia Markers, Mitochondrial Density, and Inflammation in the Liver Tissue

Liver tissue previously fixed in PFA 4% was dehydrated and embedded into paraffin. The paraffin tissue blocks were then cut into longitudinal sections of 8–10 µm thickness and transferred to slides either to perform the hematoxylin & eosin (H&E) staining protocol, for histological analysis of lipid deposition, or to evaluate by fluorescent immunohistochemistry the levels of HIF-1α, HIF-2α, F4/80 (a macrophage marker), and Mitotracker (a mitochondrial density marker).

For the histological analysis, the stained slices were visualized in a Widefield Fluorescence Microscope Zeiss Z2 for the evaluation of fat deposition.

For the immunohistochemistry protocol, the slides underwent a deparaffinization protocol and were rehydrated with ethanol and distilled water. After removing the paraffin from the slides, antigen retrieval was carried out using 0.1 M citrate buffer (pH = 6.0) for 20 min, followed by 10 min in 0.1 M glycine and 15 min of permeabilization with washing buffer (1% PBS, 0.1% Triton). Blockage of unspecific bounds was performed using a 5% Bovine Serum Albumin (BSA) in washing buffer solution for 2 h for the Mitotracker assay or a 1% BSA, 1% goat serum in washing buffer solution for the evaluation of HIF-1α and HIF-2α levels.

To evaluate HIF-1α, HIF-2α levels and F4/80, the slides were incubated overnight, at 4 °C, with the primary antibody goat anti-HIF-1α (1:100, SicGen, Cantanhede, Portugal), mouse anti-HIF- 2α (1:100, Abcam, Cambridge, UK), and mouse anti-F4/80 (1:500, SantaCruz Biotechnology, Heidelberg, Germany). After washing, the slices were incubated with secondary antibody donkey anti-goat Alexa Fluor 546 (1:5000, Abcam, Cambridge, UK) for HIF-1α, goat anti—mouse Alexa Fluor 550 (1:5000, Abcam, Cambridge, UK) for HIF-2α and donkey anti-mouse Alexa 488 (1:4000, Invitrogen, Porto Salvo, Portugal) together with 4′,6-dia-midino-2-phenylindole (DAPI) (1 mg/mL), a nuclear cell marker at room temperature protected from light. After washing, slides were mounted with coverslips and Vectashield mounting medium.

To assess the mitochondrial density, slides were incubated with Mitotracker solution (InvitrogenTM, Carlsbad, CA, USA), along with DAPI (1 mg/mL). The slides were then washed and mounted as previously described.

Negative controls were similarly incubated but in the absence of each primary antibody. The sections were visualized under the microscope (Widefield Fluorescence Microscope, Zeiss, Z2, Madrid, Spain), and the fluorescence intensity of each slide was measured using ImageJ software (https://imagej.net/software/fiji/) [20] and used to calculate the relative intensity for the area per number of nuclei determined by DAPI staining.

### 2.4. Assessment of Mitochondrial Complexes’ Enzymatic Activity

The mitochondria-rich fraction was isolated from liver tissue homogenates. In brief, using a glass-Teflon Potter Elvejhem 100 mg of liver tissue was homogenized in ice-cold potassium buffer (100 mM KH_2_PO_4_, 5 mM EDTA, pH = 7.4). The homogenate was transferred to an Eppendorf, combined with an isolation buffer (250 mM Sucrose, 10 mM HEPES, pH = 7.2) in a 1:1 ratio and centrifuged. The supernatant was collected and again centrifuged at 12,000× *g* for 15 min at 4 °C. Lastly, the final supernatant, the mitochondria-free cytosolic fraction, was collected and stored at −20 °C for further analysis. The pellet, corresponding to the mitochondria-rich fraction, was resuspended in an appropriate buffer (100 mM Sucrose, 100 mM KCl, 2 mM KH_2_PO_4_, 10 mM EDTA, pH = 7.4). The respective protein concentration was measured using the PierceTM BCA Protein Assay Kit (Thermo Fisher Scientific; Waltham, MA, USA). The mitochondria-rich fraction (10 µg of protein) was then used to assess the activity of mitochondrial respiratory complexes I and II and the citrate synthase activity. 

The citrate synthase activity was evaluated according to the protocol by Crisóstomo et al. [21], with a few minor modifications for liver tissue.

The assessment of complexes I and II activities was also based on the described protocol [21], with some modifications. Briefly, for assessing complex I activity by measuring the rate of NADH oxidation, the mitochondria-rich fraction was added to a reaction mix of a potassium phosphate buffer (25 mM KH_2_PO_4_, 5 mM MgCl_2_, pH = 7.5) supplemented with inhibitors for complex IV (0.98 mM KCN) and III (3.90 μM antimycin A); 0.19 mM decylubiquinone, an analogue of ubiquinone; and with or without 9.8 μM of rotenone, a specific inhibitor of complex I. The reaction was initiated by the addition of a 5 mM NADH solution to the mix, and the respective decrease in NADH fluorescence intensity was measured at 450 nm (λex = 366 nm) in 30 s intervals for 20 min. Complex I activity was determined by calculating the rate of NADH oxidation in the presence or absence of rotenone and expressed as relative fluorescence units (RFU), per minute and per milligram of protein. To evaluate complex II activity, the reduction rate of 2,6-dichlorophenolindo-phenol (DCPIP) was spectrophotometrically determined (ε = 20.7 mM^−1^·cm^−1^). The mitochondria-rich fraction was added to a potassium phosphate buffer (25 mM KH_2_PO_4_, pH = 7.5), supplemented with 0.98 mM KCN, 3.90 μM antimycin A, 9.8 μM rotenone, 0.19 mM decylubiquinone, and with or without 9.8 mM of oxaloacetate (complex II inhibitor). The reaction was initiated by adding a 100 mM succinate solution, and the time-dependent decrease in absorbance intensity at 600 nm was measured in 30 s intervals for 20 min. The difference in the DCIP reduction rate in the presence or absence of oxaloacetate was used to gauge complex II activity, expressed as a concentration of reduced DCPIP, per minute and per milligram of protein.

### 2.5. Determination of Intracellular ROS Content, Lipid Peroxidation, and Antioxidant Capacity of Liver Tissue

To determine the ROS levels in the liver tissue, the fluorogenic dye CM-H2DCFDA (C6827, InvitrogenTM, Carlsbad, CA, USA) was used. Briefly, 100 mg of tissue was macerated in ice-cold Tris-HCl buffer (40 mM, pH = 7.5). The homogenates were centrifuged to remove cell debris, and the supernatants were collected. Afterwards, supernatants were incubated with 10 μM CM-H2DCFDA solution for 40 min, at 37 °C, and protected from the light. Also, supernatants samples were incubated with Tris-HCl under the same conditions, as a control of the tissue autofluorescence. After, the fluorescence intensity of the samples and blanks was measured (λex = 495 nm and λem = 529 nm) using a SynergyTM H1 multi-mode microplate reader (Agilent, Santa Clara, CA, USA). Each sample was measured in duplicate, and the results are expressed as fluorescence intensity/mg protein. The protein concentration was previously assessed using PierceTM BCA Protein Assay Kit (Thermo Fisher Scientific; Waltham, MA, USA).

For the quantification of the lipid peroxide levels, the liver was homogenized in 20 mM Tris-HCl Buffer (pH 7.4; 0.2 *w*/*v*) and treated with 10 μL/mL of 0.5 M butylated hydroxytoluene in acetonitrile to prevent oxidation. LPO levels were determined in supernatant immediately after homogenization using a commercial kit and following the supplier’s instructions (Bioxytech LPO-586 kit; Oxis Health Products, Portland, OR, USA). In this assay, malonaldehyde and 4-hydroxyalkenals react with a chromogen reagent at a low temperature, yielding a stable chromophore (peak absorbance at 586 nm) and providing a reliable index of lipid peroxidation [18].

Antioxidant capacity of the liver tissue was assessed using the ferric reducing-antioxidant power (FRAP) assay that measures the tissue capability to reduce iron (III) to the ferrous form iron (II) in an acid medium. The colorimetric method was carried out using a modified method of Benzie & Strain [22]. Briefly, tissue homogenates were prepared as stated previously for ROS content determination. Subsequently, FRAP reagent, a combination of acetate buffer of 300 mM, pH = 3.6, 2,4,6-tripyridyl-s-triazine (TPTZ) 10 mM, and FeCl_3_·6 H_2_O 20 mM solutions in a 10:1:1 ratio, was used, and the reduction in the Fe^3+^–TPTZ complex to Fe^2+^–TPTZ (blue-colored complex) was measured, by monitoring the alterations in absorbance at 595 nm. The differences in the absorbance of the samples were then used to calculate the antioxidant potential values expressed as mmol of antioxidant potential/mg protein.

The antioxidant ability of liver tissue was also measured by assessing the cysteine-related thiolome of liver tissue through the quantification of the total and free total fractions of cysteine (Cys), cysteine-glycine (CysGly), and glutathione (GSH) with high-pressure liquid chromatography (HPLC) with fluorescence detection, as previously described by Correia et al. [23]. Also, the protein-bound fraction of each moiety was calculated by subtracting the total free fraction from the total fraction.

Briefly, l50 mg of liver tissue was sonicated in ice-cold phosphate buffer saline 1× (PBS1×) at 50% intensity for 10 s. Samples were centrifuged, and supernatant was collected for total and free total fractions measurements. To collect the total fraction, primarily, the sulfhydryl groups were reduced with Tris(2-carboxyethyl)phosphine hydrochloride (TCEP) solution (100 g/L, Sigma Aldrich, St. Louis, MO, USA). For protein precipitation, the samples were treated with a Trichloroacetic acid (TCA) solution (100 g/L, Carl Roth, Germany), containing 1 mM Ethylenediaminetetraacetic acid (EDTA), (Sigma Aldrich, St. Louis, MO, USA), vortexed, and centrifuged (13,000× *g*, 10 min, at 4 °C). Finally, the sulfhydryl groups were derivatized by transferring the final supernatant to 1.55 M NaOH and Na_2_B_4_O_7_·10H_2_O (125 mM with 4 mM EDTA, pH 9.5, VWR, Radnor, PA, USA) solutions, and incubated with 7-Fluoroben- zofurazan-4-sulfonic acid ammonium salt (SBD-F) solution (1 g/L, Sigma Aldrich, St. Louis, MO, USA) for 1 h, at 60 °C, and protected from the light. For the total free fraction, the samples were first submitted to the protein precipitation protocol as described above. Subsequently, the resulting supernatant was collected and incubated with the TCEP solution in the same conditions described for the total fraction. Following this incubation, the same derivatization protocol was performed.

To separate the different thiol groups, a reversed-Phase C18 LiChroCART 250-4 column (LiChrospher 100 RP-18, 5 μm, VWR, Radnor, PA, USA) was used, in a column oven at 29 °C on the isocratic elution mode for 20 min, at a flow rate of 0.8 mL/min. The mobile phase was constituted by a 100 mM acetate buffer (pH 4.5) and methanol (99:1 (*v*/*v*)). The RF 10AXL fluorescence detector was used for the detection, setting the excitation and emission wavelengths to 385 and 515 nm, respectively.

### 2.6. Statistical Analysis

Data were evaluated using GraphPad Prism Software 8.1.1 (GraphPad Software Inc., San Diego, CA, USA) and presented as mean values with SEM. Data were excluded from data set based on setting the lower limit to three standard deviations below the mean (μ − 3 × σ) and the upper limit to three standard deviations above the mean (μ + 3 × σ). Any data point falling outside this range was excluded from the data set. The significance of the differences was calculated by Two-way Analysis of Variance (ANOVA) with Bonferroni and/or Sidak’s multiple comparison test. Differences were considered significant at *p* ≤ 0.05.

## 3. Results

### 3.1. Impact of Chronic Intermittent Hypoxia on Metabolic Parameters

As expected, and as previously described [16,18], the CIH and HF diet promoted alterations in whole-body insulin sensitivity and glucose tolerance, and were herein evaluated through the HOMA-IR index (Figure 2A) and ipGTT (Figure 2B), respectively. The HF-diet-fed animals exhibited increased insulin resistance by 205.4% (HOMA-IR CTL = 6.2 ± 0.91; HF = 18.9 ± 1.92) (Figure 2A). Exposure to the CIH protocol increased insulin resistance by 94.7% in controls (CTL = 6.2 ± 0.91; CIH = 12.0 ± 1.65) and by 130.2% in HF animals (CTL = 6.2 ± 0.91; HFIH = 14.2 ± 2.37) compared to the CTL group. Also, HF diet promoted a slight increase in fasting glycemia (CTL = 122.8 ± 13.90 mg/dL; HF = 134.9 ± 21.73 mg/dL) although not statistically significant, CIH did not change this parameter in both control and obese groups. From the glucose excursion curves and from the AUC values (Figure 2B,C), it is clear that the HF diet increased glucose intolerance as the AUC of the glucose excursion curves increased significantly by 114.3% in comparison to the CTL group (CTL = 21,285 ± 2823 mg dL^−1^ min; HF = 45,611 ± 3906 mg dL^−1^ min) (Figure 2B,C). CIH increased glucose intolerance in both controls and obese animals, promoting an increase in the AUC values of 40.7% (*p*-value = 0.43) and of 138.8%, respectively (CTL = 21,285 ± 2823 mg dL^−1^ min; CIH = 29,939 mg dL^−1^ min; HFIH = 50,831 ± 5046 mg dL^−1^ min) (Figure 2B,C).

Figure 2D shows the effect of CIH and of HF diet on lipid deposition and tissue morphology assessed through the staining of H&E staining. It is clear from the analysis of the figure that CIH itself did not alter lipid deposition and tissue morphology in comparison with control animals. Furthermore, and as expected, the animals fed with a HF diet exhibited an increased number of lipid droplets in the liver, an effect that was exacerbated in obese animals submitted to CIH (Figure 2D).

In conclusion, the CIH and HF diet induce a dysmetabolic state characterized by insulin resistance and glucose intolerance.

### 3.2. Effects of Chronic Intermittent Hypoxia on Hypoxia Markers

The effects of the CIH and HF diet on the levels of two hypoxic markers, HIF-1α and HIF-2α, respectively, are displayed in Figure 3. CIH increased by 33% in the levels of HIF-1α in the liver. However, the HF diet combined with CIH significantly decreased HIF-1α levels when compared to CIH alone (CIH = 132.88 ± 6.36%, HFIH = 94.16 ± 6.21%, *p* < 0.05). Moreover, HF diet decreased by 29% HIF-2α levels in the liver, an effect not altered when the HF diet is combined with the CIH (HFIH group). Representative immunohistochemistry images of each hypoxia marker are shown in panels C and F and the corresponding fluorescence intensity quantifications are represented in panels B and E. HF diet consumption did not change the levels of HIF-1α in the liver (Figure 3B). CIH significantly increased HIF-1α levels in CTL animals by 22.2% (CTL = 35.6 ± 2.50%, CIH = 57.5 ± 1.80%, *p* < 0.001). HFIH animals showed a significant decrease in HIF-1α levels when compared to CIH group (CIH = 57.5 ± 1.80%, HFIH = 24.7 ± 2.81%; *p* < 0.0001). In contrast, regarding HIF-2α levels, 12 weeks of HF diet promoted a decrease of 9.4% (CTL = 41.7 ± 1.95%, HF = 32.3 ± 1.50%, *p* < 0.01). CIH did not alter the levels of HIF-2α in CTL animals; however, when HF is combined with CIH, we have shown a significant decrease in HIF-2α levels when compared to CIH alone (CIH = 46.7 ± 2.04%, HFIH = 24.9 ± 4.59%, *p* < 0.001). In conclusion, the CIH and HF diet applied alone differently affect the levels of HIF-1α and HIF-2α, but when the CIH and HF diet were applied together, it seems that the HF diet blunts the effect of CIH on increasing HIF-1α in the liver.

### 3.3. Effects of Chronic Intermittent Hypoxia on the Levels of Proteins Involved in Insulin Signalling and Glucose Uptake in the Liver

The HF diet promoted a significant increase in IR levels by 58.67% (CTL = 100 ± 6.28%, HF = 158.7 ± 23.42%, *p* < 0.05), whereas CIH exposure did not alter the levels of the receptor in both groups (Figure 4A). AKT levels, a downstream protein activated after insulin binding to its receptor, and a protein with a central role in the regulation of hepatic glucose and lipid metabolism [24], was not changed, either with the CIH or HF diet (Figure 4B). One important pathway for the control of plasma insulin levels and, consequently, insulin sensitivity, is insulin clearance. Therefore, we evaluated the levels of IDE, the principal enzyme responsible for insulin clearance. IDE levels did not change with the CIH or HF diet (Figure 4C). Glut2 levels, the main glucose transporter in liver tissue (Figure 4D) [25], were significantly increased by CIH in 78.1% (CTL = 100 ± 4.01%, CIH = 178 ± 32.53%, *p* < 0.05), while HF did not change Glut2 levels in control or CIH conditions. In conclusion, the whole body dysmetabolism in the HF diet and CIH animals is associated, respectively, with increased levels of the insulin receptor and of Glut2 in the liver.

### 3.4. Effects of Chronic Intermittent Hypoxia on Mitochondrial Biogenesis and Activity

We have examined the impact of CIH and/or HF diet on liver mitochondrial density, as represented in Figure 5A,B by Mitotracker labelling and correspondent fluorescent intensity quantification. HF animals exhibited a significant decrease of 12.8% (CTL = 37.7 ± 2.34%, HF = 24.9 ± 0.84%, *p* < 0.05) in mitochondrial density. CIH did not alter mitochondrial density in CTL but restored the decrease promoted by the HF diet to CTL values (HF = 24.9 ± 0.84%, HFIH = 38.3 ± 4.44%, *p* < 0.05).

To assess mitochondrial function, the levels of OXPHOS complexes were measured and represented in Figure 5C. The HF diet significantly increased complex I levels by 105.7% (CTL = 100 ± 13.56, HF = 205.7 ± 22.79%, *p* < 0.01). There was also a tendency to an increase in complexes II and IV levels promoted by the HF diet. CIH significantly decreased the levels of OXPHOS complexes I, II, IV, and V by 61.33% (CTL = 100 ± 13.56%, CIH = 38.7 ± 12.04%, *p* < 0.05), 60.8% (CTL = 100 ± 17.67%, CIH = 39.3 ± 10.49%, *p* < 0.05), 59.4% (CTL = 100 ± 12.46%, CIH = 40.6 ± 11.25%, *p* < 0.05), and 64.0% (CTL = 100 ± 13.68%, CIH = 36.0 ± 7.96%, *p* < 0.01) in CTL animals. CIH also significantly decreased by 132.9% (HF = 205.7 ± 22.79%, HFIH = 72.8 ± 6.55%, *p* < 0.001), 104.9% (HF = 151.2 ± 25.14%, HFIH = 46.2 ± 4.95%, *p* < 0.01), and 41.2% (HF = 67.9 ± 8.66%, HFIH = 26.7 ± 8.19%, *p* < 0.05) the levels of mitochondrial complexes I, II and III, respectively, in HF conditions, with a tendency to decrease mitochondrial complexes IV by 58.2% (HF = 123.7 ± 34.68%, HFIH = 65.5 ± 8.90%) and V by 70.2% (HF = 105.6 ± 34.95%, HFIH = 35.4 ± 9.16%) in the HF animals (Figure 5C).

In Figure 5D,F, the citrate synthase, mitochondrial complex I, and mitochondrial complex II activities are represented. The citrate synthase activity was used to normalize the enzymatic activity of complexes I and II with the aim to rule out reputed alterations due to different mitochondrial content between the lysates from each group [26].

No differences were observed in citrate synthase activity (Figure 5D). CIH promoted an increase in the enzymatic activity of mitochondrial complexes I in 142.4% (CTL = 2.0 × 10^5^ ± 6.41 × 10^4^ RFU, CIH = 1.9 × 10^6^ ± 8.40 × 10^5^ RFU) (Figure 5E) and the mitochondrial complex II by 149.9% (CTL = 1130 ± 318.4, CIH = 2823 ± 1063 μM DCIP) (Figure 5E,F). Both HF and HFIH groups did not show any alterations in the activity of both complexes comparing to the CTL group. In conclusion, we show that the CIH and HF diet are associated with mitochondrial dysfunction.

### 3.5. Effects of Chronic Intermittent Hypoxia on ROS Levels and on Antioxidant Capacity

The labelling with CM-H2DCFDA was carried out to measure the quantity of ROS in liver tissue, as represented in Figure 6A. CIH promoted an elevation by 31.1% on hepatic ROS levels when comparing to CTL animals (CTL = 2878 ± 365.6, CIH = 3773 ± 264.0, *p* < 0.01), an effect reverted when HF diet is also present (HFIH = 2131 ± 88.61). HF diet itself promoted a significant decrease by 52.02% (CTL = 2878 ± 365.6, HF = 1381 ± 86.11, *p* < 0.01), on hepatic ROS levels.

We also evaluated hepatic antioxidant capacity by the FRAP assay (Figure 6B) and by measuring the levels of catalase (Figure 6C), SOD-1 (Figure 6D), and cysteine-related thiolome (Figure 6E–G). In the FRAP assay, we have shown that HF diet decreased the overall antioxidant capacity significantly by 22.5% (CTL = 16.5 ± 1.20, HF = 12.8 ± 1.01, *p* < 0.05). CIH did not modify the antioxidant capacity, either in controls or in HF animals (CIH = 17.1 ± 0.70, HFIH = 13.5 ± 1.02) (Figure 6B).

The levels of lipid peroxidation given by the index of LPO in liver tissue were increased by the HF diet by 104% (CTL = 28.3 ± 0.62, HF = 58.0 ± 7.3, *p* < 0.05) without significant alterations when animals were submitted to CIH in the presence or absence of the HF diet (Figure 6C).

The levels of catalase (Figure 6D) were tendentially increased in response to HF diet (CTL = 100 ± 4.27%, HF = 147.6 ± 22.99%, *p* = 0.06), however without reaching statistical significance. CIH did not modify catalase levels, either in controls or in HF animals. In relation to SOD-1, both the CIH and HF diet induced a reduction by 62.5% and 30.4%, respectively (CTL = 100 ± 5.15%, CIH = 37.5 ± 10.79, *p* < 0.001, HF = 69.6 ± 10.20%, *p* < 0.05), an effect completely abolished in the HFIH animals (HFIH = 76.3 ± 9.77%) (Figure 6E).

When analyzing cysteine-related thiolome (Figure 6F–H), we have found that CIH produced a slight decrease in all Cys fractions (Figure 6F) in CTL animals. The HF diet induced a decrease in Cys levels in all fractions, namely by 30% (CTL= 5.6 ± 0.79 μM/mg tissue, HF = 3.9 ± 0.21 μM/mg tissue; *p*- value = 0.060) in the total fraction (left panel, Figure 6E), by 27% (CTL = 4.8 ± 0.56 μM/mg tissue, HF = 3.1 ± 0.17 μM/mg tissue; *p*-value = 0.059) in the free total fraction (middle panel, Figure 6F), and by 53% (CTL = 0.82 ± 0.16 μM/mg tissue, HF = 0.38 ± 0.09 μM/mg tissue; *p*-value = 0.044) in the protein-bound fraction (right panel, Figure 6F). The alterations in all Cys fractions in the HF diet group were not observed when combined with CIH.

Regarding total and free total fractions of CysGly (Figure 6G, left and middle panels) we have shown a 15% decrease promoted by the HF diet (CTL = 0.1 ± 0.003 μM/mg tissue, HF = 0.09 ± 0.003 μM/mg tissue, *p* < 0.01) and 26.1% (CTL = 0.038 ± 0.003 μM/mg tissue, HF = 0.028 ± 0.002 μM/mg tissue, *p* < 0.05), respectively. This effect is not altered by CIH in all CysGly fractions. In the protein-bound CysGly fraction (Figure 6G right panel), there were no alterations between the four groups. Finally, for GSH levels (Figure 6H) in CTL animals submitted to CIH, we have found a tendency for an increase in this thiol in all fractions, although without statistical significance. The HF diet in both CTL and CIH conditions did not alter GSH fractions (Figure 6H). Altogether, we can state that the whole-body dysmetabolism induced by CIH and HF diet are associated with an oxidative status in the liver.

### 3.6. Effects of Chronic Intermittent Hypoxia on Inflammatory Markers

We evaluated the effect of the CIH and HF diet on arginase I, NF-κB, IL-6R, IL-1R, TNF-α, and TNF-αR protein levels (Figure 7). Neither the CIH nor HF diet altered the levels of arginase I (Figure 7A), NF-κB (Figure 7B), and IL-6R (Figure 7C).

IL-1R (Figure 7D) was significantly increased in CTL animals exposed to hypoxia by 22.33% (CTL = 100 ± 3.80%, CIH = 122.3 ± 8.60%, *p* < 0.05). The HF diet promoted an increase in IL-1R by 32.7% (CTL = 100 ± 3.80%, HF = 132.7 ± 7.69%, *p* < 0.01).

Exposure to intermittent hypoxia in HF animals did not change IL-1R levels above the levels increased by the HF diet. The HF diet significantly increased the levels of TNF-α (Figure 7E) by 45.2% (CTL = 100 ± 7.19%, HF = 145.2 ± 13.81%, *p* < 0.05), an effect maintained when animals are also submitted to CIH (CIH = 87.8 ± 9.33%, HFIH = 141.1 ± 17.57%, *p* < 0.05). Finally, only the HF diet combined with CIH promoted a significant increase in TNF-αR levels (Figure 7F) by 35.7% (CTL = 100 ± 3.58%, HFIH = 135.7 ± 12.95%, *p* < 0.05) without any alterations in the other groups studied. Moreover, we evaluated F4/80 staining (Figure 7G,H), a marker for macrophages, within the liver and observed that CIH significantly increases the staining of this protein in control- and HF-diet-fed animals by 64.5% and 60.8%, concluding that whole body metabolic dysfunction in CIH is associated with hepatic inflammation.

## 4. Discussion

In this study, we have demonstrated that: (1) The CIH and HF diet induce insulin resistance and glucose intolerance, aggravates lipid deposition in the liver tissue, and affects HIF-1α and HIF-2α levels in the liver differently; (2) the CIH and HF diet alter mitochondrial bioenergetics, with the intake of the HF diet decreasing the mitochondrial density in the liver tissue and CIH leading to a decrease in all OXPHOS complex levels in both CTL and HF animals, increasing the enzymatic activity of complexes I and II in the CTL animals; (3) on the one hand, the HF diet decreased the levels of ROS in liver tissue, decreased the overall liver tissue antioxidant capacity, increased hepatic lipid peroxidation, and decreased both Cys (GSH precursor) and CysGly (GSH catabolic product) without changing GSH levels independently of CIH; on the other hand, CIH decreased the levels of SOD-1 and alters in different ways the levels of cysteine-related thiols in the liver; (4) CIH increased the inflammation in CTL and HF rats. As a whole, we demonstrate that mitochondrial dysfunction, oxidative stress, and inflammation are involved in the genesis of CIH-induced dysmetabolism.

As expected, the insulin signaling and the glycemic control, measured in vivo by the HOMA-IR and the ipGTT test, is impacted by both the CIH and HF diets [16]. As previously observed, the HF feeding protocol caused an increase in body weight [16,18]. Also, as previously reported, the CIH protocol herein used was also effective in altering glucose homeostasis, promoting both insulin resistance and glucose intolerance [18,26]. The exposure to CIH induced a systemic state of insulin resistance, as shown by an increase in the HOMA-IR index (Figure 2A). This is in accordance with the results obtained previously in both animal [18], and human studies [27,28]. In the HF group, we also observed insulin resistance and glucose intolerance, as previously reported in many studies. Interestingly, we found that 2 weeks of CIH exposure in animals submitted to the HF diet (HFIH group) did not exacerbate the dysmetabolic state. Several factors can contribute to this absence of exacerbation, including: the short period of exposure to CIH (only 2 weeks); the increase in lipid metabolism induced by CIH and the consequent decrease in adipocyte size, reformulation of adipose tissue function, and consequent compensatory improvement in insulin sensitivity, as demonstrated by Martins et al. [16]; or even a possible balance between hypoxia, angiogenesis, and inflammation. It is consensual that high-fat diet promotes hypoxia and inflammation; however, it is also known that hypoxia activates angiogenesis [29]. Knowing that in adipose tissue, increased angiogenesis attenuates the negative effects of hypoxia (for a review, see [30]) and that tumor hypoxia is alleviated by vascular normalization (for a review, see [31]), we can postulate that activated hepatic angiogenesis in HF-diet-fed animals may prevent the exacerbation of dysmetabolic states with 2 weeks of CIH.

We also evaluated Glut2, the main hepatic glucose transporter in rodents and humans, which was increased by CIH. This is in accordance with the results obtained by different authors [32,33], where this increase in Glut2 was described to be a compensatory mechanism for the increase in insulin resistance. Moreover, IR levels were increased by the HF diet, also showing a possible compensatory mechanism to overcome the hepatic insulin resistance promoted by this stimulus [34].

Hepatic steatosis, characterized by many hallmarks in which increased liver lipid deposition is included, is a state that is associated with obesity and T2D and that has been massively associated with OSA [5]. Herein, we showed that the hepatic lipid deposition was increased in the animals fed with the HF diet, as expected [13,35], and that the exposition to CIH aggravate this phenotype. In fact, this effect promoted by CIH has been demonstrated in human patients with OSA and obesity [15,36]. However, CIH itself did not seem to alter hepatic lipid deposition, a result that goes against previous studies with the same or a similar protocol of CIH [18,37]. These studies in comparison to our study differ in the methodology used to evaluate the hepatic steatosis; we used a qualitative evaluation by visualizing the histological staining of the liver tissue structures, instead of the quantitative analysis used in the studies described.

Trying to evaluate the effect of CIH and/or HF on insulin metabolism pathways, we assessed the levels of the main enzyme responsible for insulin clearance, IDE. Hepatic IDE levels were not affected by CIH. In the work of Minchenko et al. [38], they observed a decrease in the levels of IDE in response to hypoxia. However, this was a model of sustained hypoxia, a cancerous cell line, and a type of cell model known to have high rates of glucose use; therefore, a decrease in insulin clearance will promote an increase in insulin availability to promote the use of glucose by these cells.

We indirectly evaluated hepatic hypoxia by measuring the levels of the hypoxia markers HIF-1α and HIF-2α in this tissue (Figure 3). The levels of HIF-1α were increased in the CIH animals, as previously observed in another study from our group in response to mild CIH [32]. It was already described that HIF-1α promotes glycolysis and regulates the expression of glucose transporters in the liver [39,40]. Therefore, as in the work of Sacramento and colleagues, we can speculate that the upregulation of hepatic HIF-1α signaling correlates with the increase in Glut2 transporters herein observed. Moreover, in this study, the levels of HIF-1α and HIF-2α were found to be reduced in the HFIH group when compared to the CIH group, with no significant changes relative to the HF group. This suggests that obesity may avoid the impact of CIH on HIF-1α and HIF-2α levels in the liver. Indeed, different studies report that hyperglycemia inhibits the expression and activity of HIFs in different tissues even in conditions of hypoxia, namely pancreatic β-cells and the retina [41,42], and that HIF-2α is usually activated slower and for a longer time [42]. As previously discussed in this manuscript, this effect could potentially result from a delicate balance between hypoxia and angiogenesis [29,30,31]. It is plausible that the lack of significant alterations in HIF levels in the HFIH group might be attributed to a compensatory angiogenic response triggered by an initial hypoxia induced by the consumption of a HF diet. This response could lead to the formation of an adequate number of blood vessels, ensuring sufficient oxygen and nutrient supply to the tissue.

When evaluating hepatic oxidative status, we demonstrated that the decrease in FRAP values described in our model, accompanied by a decrease in SOD-1 levels of HF and HFIH animals, is indicative of oxidative damage which agrees with the available literature [18,43]. Nevertheless, contrary to the results obtained by Olea et al. [18] and Quintero et al. [44], we observe a decrease in SOD-1 levels in control animals exposed to CIH. Previous works related with airways exposure to hypoxia and effects on SOD activity have shown the same result as us; authors speculate that the decrease with hypoxia is consistent with an important role for SOD in protecting against cellular O2 toxicity [45].

The increase in LPO levels is restricted to liver tissue. A liver-specific upregulated expression and activation of NADPH oxidase by CIH was previously demonstrated, with a link between NADPH oxidase-derived ROS and lipid peroxidation also being suggested [44]. However, the CIH-induced LPO has been described to be CIH-severity-dependent; therefore, the absence of alterations in LPO may be due to the middle severity of our CIH paradigm. [46]. Additionally, the HF-diet-induced LPO increase was described in previous studies [46], correlating with hepatic steatosis shown also herein in our HF diet animals, and with a decrease in antioxidant capacity and mitochondrial damage.

Moreover, the levels of glutathione, the principal non-enzymatic antioxidant defense, were not altered by CIH, which is in agreement with the study of Quintero et al. [44]. Also, the HF diet produced a decrease in both Cys (precursor of GSH) and CysGly (catabolite of GSH), independently of CIH concomitancy. Altogether, this might be indicative of a lower degradation of glutathione in conditions of HF diet to prevent oxidative stress. Also, the Cys decrease might represent an increase in Cys metabolism. Cysteine in metabolic disease goes beyond antioxidant properties; namely, Cys metabolism controls the toxicity of cysteine excess and is a pivotal source of metabolites to biomass and bioenergetics, as well as a promoter of adaptive responses to stressors [47,48]. For instance, the Cys involvement in the adaptation to sustained hypoxia by promoting metabolic reprogramming in order to cope with a high metabolic demand and challenging oxidative stress conditions has been increasingly recognized [48]. Overall, these results indicate an increase in oxidative stress in response to the CIH and HF diet.

Since mitochondrial dysfunction is related to the occurrence and development of various chronic liver diseases, including metabolic diseases, is also related to OSA and sleep-related diseases, we evaluated mitochondrial biogenesis and activity in hepatic tissue. We showed that the number of mitochondria in the liver tissue is reduced in response to the HF diet, which is manifested as a decrease in Mitotracker levels. Also, the levels of OXPHOS are reduced in response to CIH both in physiologic and hypercaloric diet conditions. In addition, we also show that the enzymatic activity of complexes I and II tends to increase in the CIH group. Chen et al. [49] also observed a reduction in the mitochondria number in the genioglossus, the largest upper airway dilator muscle whose function is altered in patients with OSA. Also, the increase in FFA, a hallmark related to the increase in hepatic steatosis, can be contributing to mitochondrial damage and, therefore, to the reduction in mitochondria levels in these conditions [13]. The decrease in OXPHOS levels in conditions of hypoxia is indicative of mitochondrial damage, probably induced by ROS. This result agrees with the available literature [50,51]. Regarding the enzymatic activity of the complexes: it is described as a reduction in complex I and II activity under hypoxic conditions [52,53]. However, these results were obtained in models of sustained hypoxia; thus, our CIH paradigm might produce the activation by different mechanisms or a compensation for the decrease in the complexes’ levels and mitochondrial function.

Finally, a low grade of inflammation is known to run with obesity and metabolic dysfunction [5,13]. We observed an increase in the levels of IL-1R, TNF-αR, and TNF-α, in cluding proinflammatory markers, confirming a proinflammatory state within the tissue. In agreement, we showed that CIH increased levels of F4/80 staining, a macrophage cell marker, in both control and HF diet animals, confirming the increased inflammation in these conditions. Knowing that F4/80 staining is high in Kupfer cells and in liver monocyte-derived macrophages (MoMFs) [54,55] and that the activation of these cells drives macrophage polarization and activation [56,57], it is plausible to conclude that CIH drives liver inflammation, contributing to hepatic and whole-body metabolic dysfunction.

Altogether, these results indicate that mitochondrial dysfunction, oxidative stress, and inflammation in the liver are involved in the genesis of CIH-induced dysmetabolism.

## 5. Conclusions

The relationship between metabolic diseases and OSA is undeniable; however, the subsequent mechanism(s) that link them is yet to be clarified. With the increase in prevalence of these disorders, deciphering the mechanisms underlying this relationship are crucial to, ultimately, discover new potential therapeutic targets and develop new therapies to allow the improvement of the metabolic burden of OSA patients. Our results suggest that hepatic mitochondrial dysfunction and oxidative stress, along with the consequent onset of inflammation, might be key factors contributing to the development of this CIH-induced dysmetabolic phenotype.

## Figures and Tables

**Figure 1 antioxidants-12-01910-f001:**
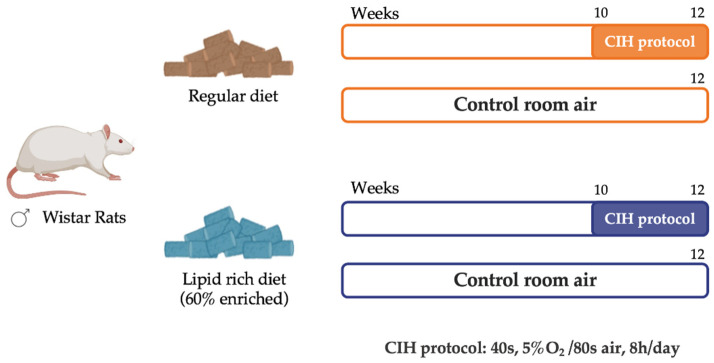
Schematic representation of the study experimental design. Male Wistar rats (12 weeks) were divided into a control group (CTL), fed with a standard diet, and an obese group (HF), fed with a 60% lipid-enriched diet for 12 weeks. From weeks 10 to 12, half of the animals from both groups were submitted to a CIH protocol (CIH and HFIH) consisting of 30 intermittent hypoxia cycles/h, 8 h/day. During the CIH protocol, the remaining age-matched control or high-fat animals were in the same room and exposed to a normal air atmosphere, to experience similar conditions.

**Figure 2 antioxidants-12-01910-f002:**
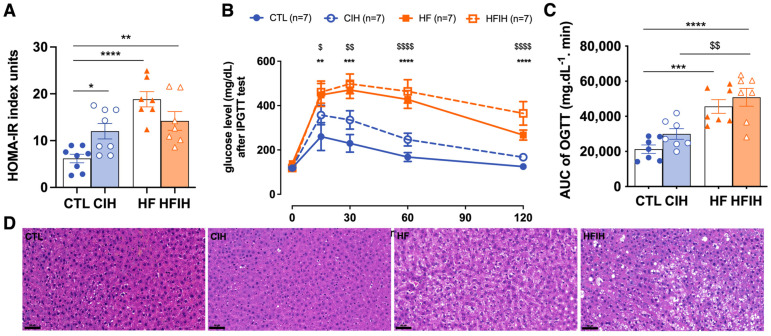
Effects of chronic intermittent hypoxia (CIH) and high-fat diet (HF) on whole-body metabolic parameters, insulin sensitivity (**A**), and glucose tolerance (**B**,**C**); and on liver lipid deposition (**D**). Panel (**A**) shows the effect of CIH and HF diet on insulin sensitivity evaluated using HOMA-IR index. Panel (**B**) shows, on the left, the glucose excursion curves of the intra-peritoneal glucose tolerance test (ipGTT) and on the right, the correspondent area under the curve (AUC). Panel (**D**) shows from the left to the right representative images of H&E staining from CTL, CIH, HF, and HFIH animals. Visual analysis of H&E staining shows an increase in lipid deposition in the HF group, which is aggravated by exposure to chronic intermittent hypoxia (Scale bar 50 μm). Animal groups: CTL-control; CIH-chronic intermittent hypoxia; HF-high-fat diet; HFIH-high-fat plus CIH; Data are presented as means ± SEM. Two-way ANOVA with Bonferroni and Sidak’s multiple comparison tests: * *p* < 0.05, ** *p* < 0.01, *** *p* < 0.001 and **** *p* < 0.0001 compared with control animals; $ *p* < 0.05, $$ *p* < 0.01, $$$$ *p* < 0.0001 compared with control animals submitted to the CIH protocol.

**Figure 3 antioxidants-12-01910-f003:**
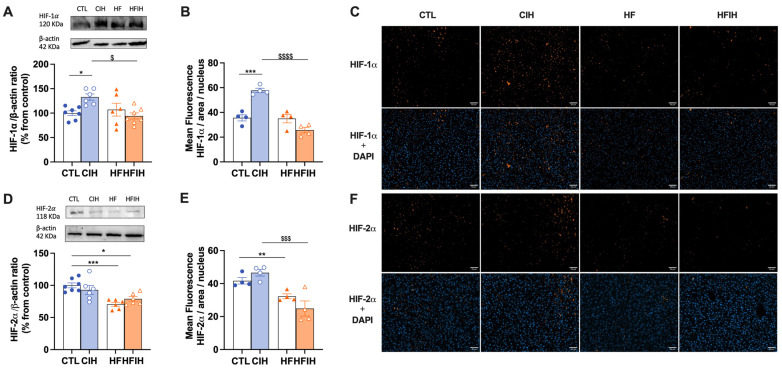
Effects of chronic intermittent hypoxia (CIH) and high-fat (HF) diet on HIF-1α (**A**–**C**) and HIF-2α (**D**–**F**) levels in the liver. Panel (**A**,**D**) depicts the levels of HIF-1α (120 kDa) and HIF-2α (118 kDa), respectively, in the liver. Protein levels were normalized to the loading control β-actin (42 kDa). On the top of each graph are presented the representative Western blots for each protein studied. Panel (**B**,**F**) show the fluorescence quantification for HIF-1α and HIF-2α (orange fluorescence), respectively, normalized for the number of nuclei present in the liver labelled with DAPI staining (blue fluoresecence). Panel (**C**,**F**) are representative immunohistochemistry images from liver labelled with HIF-1α and HIF-2α, respectively, and the merged images with DAPI for each hypoxia marker (scale bar: 50 μm). Animal groups: CTL-control; CIH-chronic intermittent hypoxia; HF-high-fat diet; HFIH-high-fat plus CIH; data are presented as means ± SEM. Two-way ANOVA with Bonferroni and Sidak’s multiple comparison tests, respectively: * *p* < 0.05, ** *p* < 0.01, *** *p* < 0.001 compared with CTL animals; $ *p* < 0.05, $$$ *p* < 0.001, $$$$ *p* < 0.0001 compared with CTL animals submitted to the CIH protocol.

**Figure 4 antioxidants-12-01910-f004:**
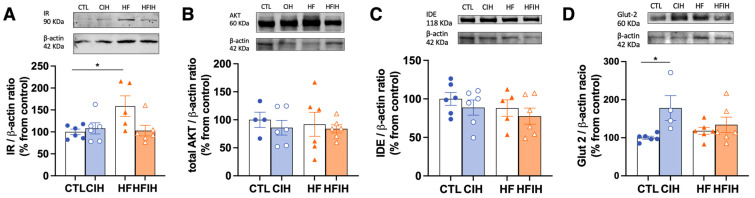
Effects of chronic intermittent hypoxia (CIH) and high-fat (HF) diet on the levels of proteins involved in insulin signaling and glucose transport in the liver. Panels (**A**–**D**) depicts the levels of insulin Receptor (IR, 90 kDa), protein kinase B (Akt, 60 kDa), insulin degrading enzyme (IDE, 118 kDa, and glucose transporter 2 (Glut2, 60 kDa), respectively. Protein levels were normalized to the loading control β-actin (42 kDa). On the top of each graph are presented the representative Western blots for each protein studied. Animal groups: CTL—control; CIH—chronic intermittent hypoxia; HF—high-fat diet; HFIH—high-fat plus CIH. Data are presented as means ± SEM. Two-way ANOVA with Bonferroni and Sidak’s multiple comparison tests, respectively: * *p* < 0.05 compared with control animals.

**Figure 5 antioxidants-12-01910-f005:**
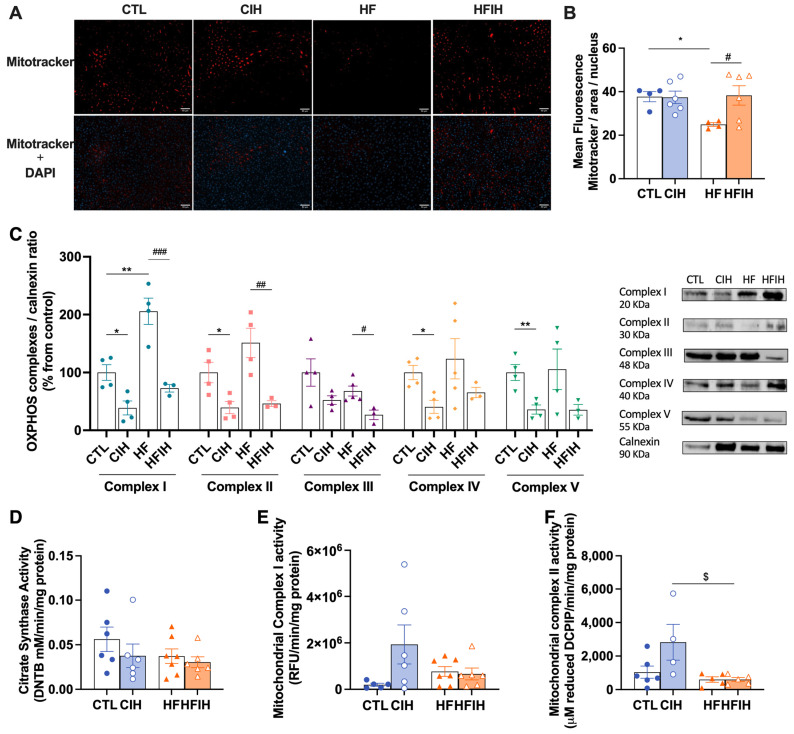
Effects of chronic intermittent hypoxia (CIH) and high-fat (HF) diet on the mitochondrial density, evaluated by Mitotracker assay; oxidative phosphorylation complexes levels, evaluated by Western blot analysis of OXPHOS complexes levels; and activities, namely citrate synthase, complex I, and complex II activity assessment. Panels (**A**,**B**) present the representative immunohistochemistry staining of Mitotracker (red fluorescence) and merge with DAPI (blue fluoresence) and the fluorescent intensity quantification, respectively (scale bar 50 μm). In panel (**C**), the graph depicts the levels of complex I (20 kDa), II (30 kDa), III (48 kDa), IV (40 kDa), and V (55 kDa). Proteins levels were normalized to the loading control, calnexin (90 KDa). On the right side of the graph, we show representative Western blots for each protein evaluated. On panel (**D**–**F**), the activity of citrate synthase, mitochondrial complex I, and mitochondrial complex II, respectively, is represented. The enzymatic activity of each complex was normalized by citrate synthase activity. Animal groups: CTL—control; CIH—chronic intermittent hypoxia; HF—high-fat diet; HFIH—high-fat plus CIH. Data are presented as means ± SEM. Two-way ANOVA with Bonferroni and Sidak’s multiple comparison tests, * *p* < 0.05, ** *p* < 0.01 compared with CTL animals, # *p* < 0.05, ## *p* < 0.01, ### *p* < 0.001 compared with HF animals, and $ *p* < 0.05 compared with CTL animals submitted to the CIH protocol.

**Figure 6 antioxidants-12-01910-f006:**
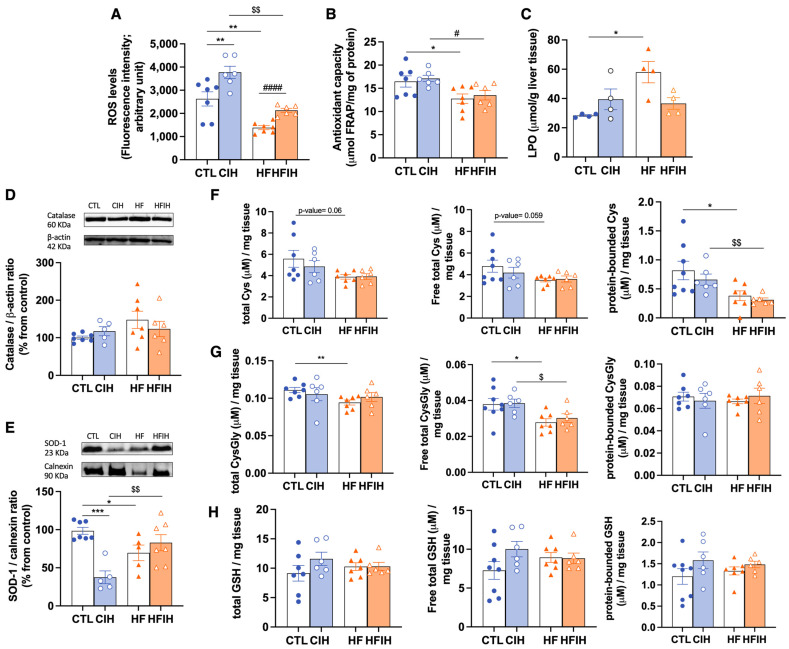
Effects of chronic intermittent hypoxia (CIH) and high-fat (HF) diet on ROS levels, lipid peroxidation (LPO), and on the antioxidant capacity of the liver. Panel (**A**) depicts the intracellular levels of ROS in the liver measured by CM-H2DCFDA labelling. Panel (**B**) presents the FRAP assay showing the overall antioxidant capacity of the tissue. Panel (**C**) depicts the levels of LPO shown by the index of lipid peroxidation measured per g of liver tissue. Panels (**D**,**E**) depicts the levels of catalase (60 KDa) and superoxide Dismutase 1 (SOD-1, 23 KDa), respectively. Protein levels were normalized to the loading control β-actin (42 kDa) and calnexin (90 KDa), respectively. Top of the graphs show representative Western blots for each protein studied. Panels (**F**–**H**) represent the levels of cysteine-related thiols in the liver, namely Cys on panel (**F**), CysGly in panel (**G**), and GSH in panel (**H**). In each of these panels, it is shown on the left the total fractions of each thiol, in the middle the free total thiol fraction, and on the right the protein-bounded thiol fraction. Animal groups: CTL—control; CIH—chronic intermittent hypoxia; HF—high-fat diet; HFIH—high-fat plus CIH. Data are presented as means ± SEM. Two-way ANOVA with Bonferroni and Sidak’s multiple comparison tests: * *p* < 0.05, ** *p* < 0.01, and *** *p* < 0.001 compared with CTL animals; # *p* < 0.05 and #### *p* < 0.0001 compared with HF animals; $ *p* < 0.05 and $$ *p* < 0.01 compared with CIH animals.

**Figure 7 antioxidants-12-01910-f007:**
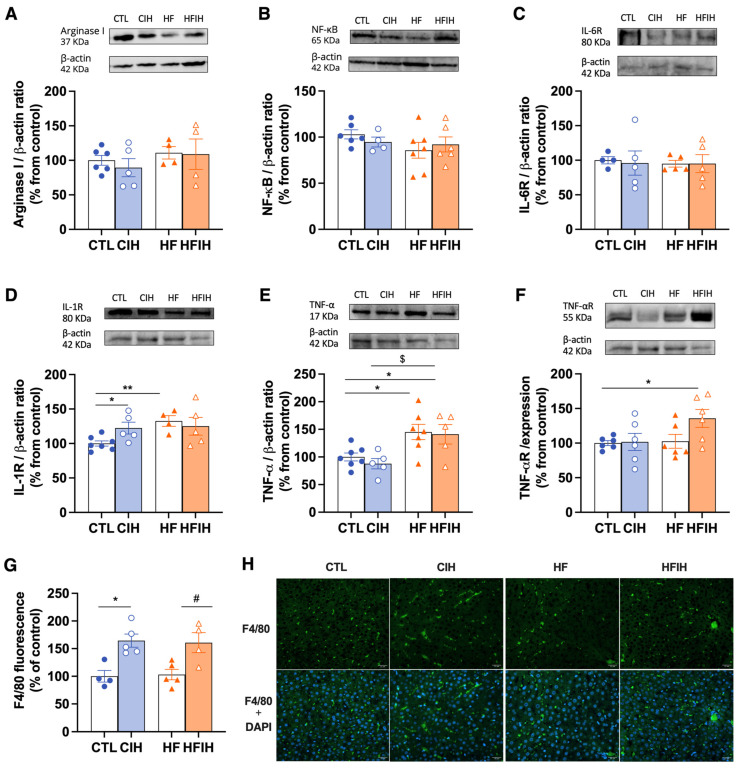
Effects of chronic intermittent hypoxia (CIH) and high-fat (HF) diet on the levels of inflammatory markers in the liver. Panel (**A**–**F**) depicts, respectively, the levels of Arginase I (35 KDa), NF-κB (65 KDa), IL-6 Receptor (IL-6R, 80 KDa), IL-1 Receptor (IL-1R, 80 KDa), tumor necrosis factor alpha (TNF-α, 17 KDa), and TNF-α Receptor (TNF-αR, 55 KDa). Protein levels were normalized to the loading control β-actin (42 kDa). Top of the graphs show representative Western blots for each protein studied. Panel (**G**,**H**) depicts, respectively, the mean fluorescence of F4/80 per area (% of control), immunofluorescence images for positive staining for F4/80 (green fluorescence, top panel), and merge with the DAPI (blue fluorescence, bottom panels) (scale bar 25 μm). Animal groups: CTL—control; CIH—chronic intermittent hypoxia; HF—high-fat diet; HFIH—high-fat plus CIH. Data are presented as means ± SEM. Two-way ANOVA with Bonferroni and Sidak’s multiple comparison tests: * *p* < 0.05 and ** *p* < 0.01 compared with CTL animals; $ *p* < 0.05 compared with CIH animals; # *p* < 0.05 compared with HF animals.

## Data Availability

Data are not publicly available due to a lack of time between data obtention and publication. The data that support the findings of this study are available from the corresponding author upon request.

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
