# Peer review of "Chronic Intermittent Hypoxia-Induced Dysmetabolism Is Associated with Hepatic Oxidative Stress, Mitochondrial Dysfunction and Inflammation"

_antioxidants, 2023, doi:10.3390/antiox12111910_

Round 1
Reviewer 1 Report
Manuscript by Fernandes et al investigates molecular basis for the chronic intermittent hypoxia induced-metabolic dysfunction. Authors used an in vivo model of high-fat diet and chronic intermittent hypoxia and interrogated multiple pathways of metabolic dysfunction. The experimental plan and study are well-done.
Major Comments:
1) Fig. 1D: Representative figure shown in HF group shows minimal steatosis after 12 weeks of HFD. A specific lipid staining (such as oil red o or fluorescent markers for lipids) and quantitation should be performed.
2) Fig. 6A: Compared with CTL, the HF diet group indicates decreased ROS levels. This also contradicts with High-fat model of Metabolic-dysfunction associated fatty liver disease (MAFLD) and should be discussed in the discussion part.
3) Fig. 7: Histopathological assessment of infiltrating inflammatory cells should be performed (e.g. F4/80 staining), as inflammatory marker genes are indicating variability.
4) Include brief conclusions (regarding role of specific pathways e.g. glucose/insulin resistance, oxidative stress/ROS, inflammation, mitochondrial function) at the end of each sub-sections within the results section to enhance the reading experience.
Author Response
We appreciate the reviewer comments and its wiliness to improve the quality of our work. We hope that in the present form the reviewer finds the manuscript suitable for publication.
Major Comments:
- 1D: Representative figure shown in HF group shows minimal steatosis after 12 weeks of HFD. A specific lipid staining (such as oil red o or fluorescent markers for lipids) and quantitation should be performed.
We appreciate the reviewer comment related with the steatosis development in the animals submitted to HF diet. It is well described that fatty liver disease liver steatosis is commonly connected with HF diet and obesity and also it was described that OSA is associated with NAFLD independently of obesity (https://doi.org/10.1164/rccm.201806-1109TR and for a review see doi: 10.1111/jsr.13418). However, in the present manuscript our aim was to study the mechanism behind the impact of chronic intermittent hypoxia in the development of insulin resistance and glucose intolerance, more than focus on liver steatosis. We believe that focus on liver steatosis is outside the scope of the present manuscript.
- 6A: Compared with CTL, the HF diet group indicates decreased ROS levels. This also contradicts with High-fat model of Metabolic-dysfunction associated fatty liver disease (MAFLD) and should be discussed in the discussion part.
The reviewer's point is very interesting. In Figure 6A, we illustrate intracellular ROS levels in the liver, measured through CM-H2DCFDA labeling. This compound, initially non-fluorescent, possesses the ability to traverse cell membranes. Once nestled within the cellular domain, CM-H2DCFDA undergoes enzymatic cleavage by intracellular esterases, metamorphosing into 5-(and-6)-chloromethyl-2',7'-dichlorodihydrofluorescein (CM-H2DCF)—a compound remarkably sensitive to oxidation. CM-H2DCF is sensitive to ROS, particularly hydrogen peroxide (H2O2) and peroxides, undergoing a transformative process that culminates in dichlorofluorescein (DCF). This is fluorescent and directly related with the intracellular ROS levels. Yet, in this pursuit of clarity, we must acknowledge the multifaceted nature of our findings. It is not a solo performance; rather, it is an orchestrated by myriad factors. The overture includes considerations of mitochondrial function and the liver's detoxication system. Our data, illustrates that control mice exhibit more efficient mitochondria and a finely tuned detoxification system. This translates into a reduction in ROS accumulation detectable by CM-H2DCFDA labeling and further supports our hypothesis. Thank you for your suggestion.
- 7: Histopathological assessment of infiltrating inflammatory cells should be performed (e.g. F4/80 staining), as inflammatory marker genes are indicating variability.
We appreciate the comment made by the reviewer and as suggested we performed some immunofluorescence analysis of liver F4/80 staining in the different experimental groups that have been included in figure 7 and discussed in the discussion section.
- Include brief conclusions (regarding role of specific pathways e.g. glucose/insulin resistance, oxidative stress/ROS, inflammation, mitochondrial function) at the end of each sub-sections within the results section to enhance the reading experience.
We have now included brief conclusions at the end of each section.
Reviewer 2 Report
It is a well-written manuscript. Here are my comments.
1. Line 121-122 “The CIH protocol consisted of 30 cycles/h 121each cycle with an exposure for 40 s to 5% O2, followed by an exposure for 80 s to air, for 122 8 h per day (apnoea-hypopnoea index of 30).”
Could authors show the actual oxygen and CO2 content in the rats exposed to CIH?
2. “2.4. Assessment of mitochondrial complexes enzymatic activity”
It is unclear why authors only measured enzyme activity in frozen tissue? The quality of manuscript will be significantly improved if authors measured mitochondrial function in freshly isolated mitochondria.
Line 237-238 “To evaluate complex II activity, the reduction rate of 2,6-dichlorophenolindo-phe- nol (DCPIP) was spectrophotometrically determined”.
It is good to know that authors used the traditional method to measure complex II activity. The same method should be used to measure complex I activity.
3. Figure 2.
There were no statistical makers in Figure 2B.
There were only 7 animals in Figure 2A HF group, but there were 8 animals in Figure 3B HF group. Please give explanation.
Figure 3
Figure 3A, there were variability in β-actin content. It is better to use the total protein content as loading control.
Molecular markers should be shown on the gel.
Figure 3A FIH group had 7 samples, but Figure 3D FIH group only had 6 samples. Why??
Figure 4
It is better to use the total protein content as loading control.
Please explain why there were different animal number in different blots.
For example, there were 6 samples in Figure 4A CTL group, but only have 4 samples in Figure 4B CTL group.
Figure 5
Again, there were issues on animal numbers.
Figure 2 showed that there were 8, 7, 8, 7, animals in CTL, CIH, HF, and HFIH groups.
The number of animals in Figure 5 did not match the number in Figure 1. Why???
If authors excluded the animals from the analysis, authors should state the criteria to exclude the outline numbers.
Figure 5 C clearly showed that HF led to decreased complex V protein content. Complex V activity should be also measured in addition to complex I and II activities.
Figure 6.
Again, please match the animal number with Figure 1 in each group.
Figure 7
There were more variabilities in β-actin content between groups. The total protein should be used as loading control.
4 line 768 “Cells 2022 11, 3735.” A comma is missed between 2022 and 11.
Author Response
We appreciate the reviewer comments and its wiliness to improve the quality of our work. We hope that in the present form the reviewer finds the manuscript suitable for publication.
Please see the answers to the comments in the documented attached.

Round 2
Reviewer 1 Report
All comments have been addressed and/or clarified by the authors.
Reviewer 2 Report
My concerns have been properly addressed in the revised manuscript.